Reproductive characteristics and gametogenic cycle of the scleractinian coral Dendrophyllia ramea

http://orcid.org/0000-0002-2580-1002 Orejas Covadonga 1 cova.orejas@ieo.csic.es
Antón-Sempere Silvia 2 3
Terrón-Sigler Alexis 4
http://orcid.org/0000-0002-6461-6985 Grau Amalia 3
1 Centro Oceanográfico de Gijón, Spanish Institute of Oceanography (IEO-CSIC) , Gijón, Asturias , Spain
2 Centro Oceanográfico de Baleares, Spanish Institute of Oceanography (IEO-CSIC) , Palma, Baleares , Spain
3 Laboratorio de Investigaciones Marinas y Acuicultura (LIMIA, IRFAP), Gobierno de las Islas Baleares , Puerto de Andratx, Baleares , Spain
4 Hombre y Territorio , Sevilla, Andalucía , Spain
Banaszak Anastazia
Electronic publication date: 2023 Sep 29
Publication date: 2023
Volume: 11
Electronic Location ID: e16079
Received 2023 Apr 4; Accepted 2023 Aug 21
Copyright: © 2023 Orejas et al.
Copyright year: 2023
Copyright holder: Orejas et al.
License: This is an open access article distributed under the terms of the Creative Commons Attribution License, which permits unrestricted use, distribution, reproduction and adaptation in any medium and for any purpose provided that it is properly attributed. For attribution, the original author(s), title, publication source (PeerJ) and either DOI or URL of the article must be cited.
License URL: https://creativecommons.org/licenses/by/4.0/

Keywords: Mediterranean, Reproduction, Gametogenic cycle, Gonochoric species, Dendrophyllia ramea

Funding: European Union’s Horizon 2020 818123 This study has received funding from the European Union’s Horizon 2020 research and innovation program under the project i Atlantic (Grant agreement No. 818123). This output reflects only the author’s view and the European Union cannot be held responsible for any use that may be made of the information contained therein. The funders had no role in study design, data collection and analysis, decision to publish, or preparation of the manuscript.

==============================
The present study marks a pioneering investigation into the reproductive cycle of the scleractinian coral Dendrophyllia ramea. This is one of the first reproduction studies conducted in the Mediterranean Sea for a colonial azooxanthellate coral. Coral samples were collected in 2017 (May and October) and 2018 (February and July) in the Alborán Sea (SW Mediterranean). This location was selected due to its rarity as one of the few sites where this species thrives at depths shallower than 40 m. These samples were used to study the sexual patterns, fertilization mechanisms and gametogenic cycles by means of histological techniques. To broaden the scope, Sea Surface Temperature (SST) and Chlorophyll-a (Chl-a) data from open access databases have been considered to explore the potential influence of these environmental factors as triggers for gamete development and spawning time. The findings cast D. ramea as a gonochoric species, since no hermaphroditic specimens were observed among the analysed samples. Additionally, the lack of larvae and embryos in any of the analysed polyps, suggest that this species is fertilised externally. Two oocyte cohorts have been detected simultaneously, hinting at a yearly reproductive cycle, characterised by a prolonged oocyte maturation and seasonal spawning period taking place between August and October. Nevertheless, D. ramea display a low fecundity compared to other scleractinians inhabiting deep waters. Lastly, the early stages of gametogenesis seem to be coupled with the highest Chl-a values (i.e., March and December), whereas spawning takes place throughout the warmest period of the year (August to October).

Introduction

The knowledge on the reproductive cycle and characteristics of azooxanthellate scleractinians, and more specifically deep and cold-water scleractinian corals, has increased over the last decade (e.g., Airi et al., 2016; Brooke & Järnegren, 2013; Feehan, Waller & Häussermann, 2019; Pires, Silva & Bastos, 2014; Prasetia et al., 2017; Shlesinger & Loya, 2016; Waller & Feehan, 2013). However, despite the fact that more than 3,000 Cold-Water Coral (CWC) species have been described so far, reproductive information is only known for less than 60 species (see Brooke & Stone 2007; Feehan & Waller, 2015; Rakka et al., 2017; Rossin, Waller & Försterra, 2017); a disparity that is particularly apparent regarding Mediterranean CWCs (Airi et al., 2016; Orejas & Jiménez, 2019). Harrison (2011), summarized the reproductive characteristics of scleractinian corals (encompassing shallow, mesophotic and CWCs), showing that 71% of scleractinian corals are hermaphrodites, whereas 26% are gonochoric species and only 3% are showing mixed patterns. Nevertheless, despite shallow-water scleractinian corals being mostly hermaphrodites (Fadlallah, 1983a; Harrison & Wallace, 1990; Richmond & Hunter, 1990), most CWC scleractinians studied to date have been reported to be gonochoric (Feehan, Waller & Häussermann, 2019; Waller, 2005; Waller & Feehan, 2013; Waller et al., 2023). The large number of hermaphroditic species found in shallow waters (which have been more investigated, mainly due to a better accessibility) originally suggested that hermaphroditism was the most ancestral reproductive condition in scleractinian corals (Szmant, 1986). However, subsequent studies pointed to gonochorism and thereby the most primitive form of reproduction (Goffredo, Arnone & Zaccanti, 2002; Harrison, 1990). Therefore, external mating is the most common fertilisation strategy for scleractinians (Harrison, 2011), even for those thriving in deep waters (Waller, 2005).

Most of the studies focusing on scleractinian reproduction, have been conducted on tropical species (Fadlallah, 1983a; Harrison & Wallace, 1990). However, reproduction studies on mesophotic (Shlesinger & Loya, 2019) and CWCs have substantially increased during the last decades (i.e., Brooke & Järnegren, 2013; Feehan, Waller & Häussermann, 2019; Larsson et al., 2014; Strömberg & Larsson, 2017; Waller, Tyler & Gage, 2002, 2008; Waller & Feehan, 2013; Waller & Tyler, 2005, 2011; Waller et al., 2023). Notwithstanding, our current knowledge on the reproductive biology of scleractinians from temperate and cold waters is still scarce, in particular regarding the Mediterranean (Goffredo et al., 2006). Indeed, reproductive data of Mediterranean corals mostly originates from observations of the species Caryophyllia smithii, Balanophyllia regia, Leptosammia pruvoti, Astroides calycularis and Cladopsammia rolandi by Lacaze-Duthiers (1873), as well as from Balanophyllia europea (Goffredo, Arnone & Zaccanti, 2002), L. pruvoti (Goffredo et al., 2005, 2006), A. calycularis (Goffredo et al., 2010, 2011, Terrón-Sigler, 2016), Cladocora caespitosa (Kružić, Žuljević & Nikolić, 2008) and Caryophyllia inornata (Caroselli et al., 2017; Goffredo et al., 2012; Marchini et al., 2020) which have been studied most recently. From these species, only C. smithii, C. inornata and L. pruvoti cover a wide bathymetric range, from shallow waters to more than 150 m in depth (Altuna & Poliseno, 2019).

Dendrophyllia ramea is a colonial azooxanthellate scleractinian coral with a Mediterranean-Atlantic distribution and a depth range from 40 to 240 m (Angiolillo et al., 2022; Brito & Ocaña, 2004; Dias et al., 2020; Kružić, Zibrowius & Pozar-Domac, 2002; Orejas et al., 2019a, 2019b; Salomidi et al., 2010; Salvati et al., 2021; Zibrowius, 1980). While certain records are documented in Pacific waters within the Ocean Biodiversity Information System (OBIS) data base (https://mapper.obis.org/?taxonid=135187), it is important to note that these could potentially be attributed to mis-identifications. The species present a remarkable morphological plasticity, with colonies displaying different shapes (Orejas et al., 2017, 2019a, 2019b). The corallite diameter spans from 5 to 12 mm, exhibiting a consistent distribution along the branches in two opposing rows (Kružić, Zibrowius & Pozar-Domac, 2002). Since their initial discovery in the Mediterranean Sea, the prevailing assumption was that this species predominantly flourished on hard substrata, encompassing a depth range of ~40 to ~60 m (Zibrowius, 1980). This pattern diverges significantly from the populations in the Canary Islands, where D. ramea tends to inhabit deeper waters (i.e., 80–150 m) and generally thrives on biogenic hard substrata (Brito & Ocaña, 2004). However, recent discoveries in the Mediterranean reveal the occurrence of this species living on soft substrata at 125–170 m depth in Cypriot waters (Orejas et al., 2019a, 2019b), with the deepest population of D. ramea occurring in the Menorca channel at 240 m depth (Jiménez et al., 2016). Furthermore, several populations of this species have been located in the Alborán Sea in depths as shallow as 16 m (Terrón-Sigler, 2016). Therefore, although most occurrences have been documented for mesophotic and deep-water environments, D. ramea displays a large bathymetric range. In order to ensure consistency with the previous publications on D. ramea from the Mediterranean (Orejas et al., 2019a, 2019b), and taking into account the species’ common occurrence in temperatures above 10–12 °C, this study has opted to utilize the term “Deep-sea coral” (DSC) to refer to this species, instead of the term “Cold-water coral” (CWC).

The primary objective of this study is to provide the first description of the sexual characteristics of Dendrophyllia ramea in the Mediterranean Sea, specifically at the coast of Granada (SW Mediterranean, Alborán Sea), as well as to quantify and describe its gametogenic cycle, and reproductive timing. The obtained results are discussed in the context of the current knowledge regarding the reproduction of scleractinian corals, including CWCs.

Materials and Methods

Sampling and study site

Samples of Dendrophyllia ramea (Fig. 1) were collected between 30 and 37 m depth, in a location inhabited by a dense population of the species (Terrón-Sigler, 2016). The latter is found off the coast of Granada, in Punta de La Mona (36°43′25″ N 3°43′56″ W, northern Alborán Sea, western Mediterranean Basin) (Figs. 2A and 2B), within the Marine Protected Area (MPA) of Acantilados y Fondos Marinos de la Punta de la Mona. This MPA was established in 2015 as a Special Area of Conservation (SAC) declared by the Autonomous Andalusian government (Junta de Andalucía) (369/2015/BOJA). The MPA is characterised by the upwelling of cold and nutrient-rich water, resulting from the anticyclonic gyres promoted by incoming Atlantic waters in the Mediterranean Sea (La Violette, 1984). Upwelling is more evident in the summer, when the thermal contrast is stronger. Although the MPA extends below 40–50 m depth, the species tends to occur in relatively shallow waters in this area (Cebrián & Ballesteros, 2004). The relatively shallow occurrence of the species in this location allowed to perform four sampling events, covering all seasons. Sampling was conducted by scuba divers in May (spring) and October (autumn) 2017, as well as in February (winter) and July (summer) in 2018 (Table 1, Supplemental Material 1). Three to five polyps were collected from each of the 38 different colonies. Sampling was conducted under the permit (001997/A04D) of the Autonomous Andalusian government (Junta de Andalucía) (Supplemental Material 1). After sampling, polyps were preserved in sea water with buffered formaldehyd with a final concentration of 4% and later transferred to the Laboratorio de Investigaciones Marinas y Acuicultura (LIMIA–IRFAP, Mallorca) for histological analyses.

Figure 1 Underwater image of a Dendrophyllia ramea colony from study area (scale bar = 5 cm).

Figure 2 Study area.

(A) Location of the study area, (B) shows the specific site where the sampling took place close to the coast of Granada Punta de la Mona (36°43´25´´N; 003° 43´56´´W) (Figure has been created by A. Terrón-Sigler).

Table 1 Dendrophyllia ramea sampling in Punta de La Mona.

	Depth (m)	Date	F	M	NI	T	
			C	P	C	P	C	P	C	P	
	36-37	27/05/17	3	9	3	3	0	0	6	12	
	34-36	02/10/17	3	9	2	2	3	6	5	11	
	36-37	03/02/18	3	9	3	3	1	1	6	12	
	33-34	29/07/18	3	9	3	3	1	1	6	12	
Total C/P			12	36	11	11	5	8	23	47	
Note:

All samples have been collected in the same sampling location (36°43´25´´N; 003°43´56´´W). Table includes depth (m), date (day/month/year) and the number of colonies per each sex. F, female colonies; M, male colonies; NI, sex non-identified; T (females and males), total number (colony/polyp); C, colony number; P, polyp number.

Biometry and histological processing

Biometric and histological analyses were performed for all (47) polyps before the decalcification process was conducted (see Table 1). For the biometric analysis, the polyp calyx diameter (D, major axis of the oral disk) and the polyp height (H, oral-aboral distance) were measured. The total number of mesenteries per polyp was counted in a total of five haphazardly selected polyps from different months. Polyp measurements (calyx diameter and polyp height) were performed in order to explore potential relationships between reproductive output and polyp size. Total polyp fecundity was then extrapolated from the number of gametes found during the histological analyses (conducted in three mesenteries). Polyp mesenteries were processed for histological analyses in order to: (1) determine the sex of each polyp (2) quantify gametes and describe their developmental stages, and (3) describe the gametogenic cycle. Polyps underwent decalcification by using a 10% formic acid solution for 24–48 h. Once the polyps were fully decalcified, each polyp was dissected and three mesenteries per polyp were extracted to perform gamete counting, analyse and describe their developmental stages and describe the gametogenic cycle. Extracted mesenteries were included in histological cassettes and subsequently dehydrated in a graded ethanol series (70–100%), cleared by using Microclearing X0026® and subsequently embedded in paraffin wax. Three embedded mesenteries of three haphazardly selected polyps from three haphazardly selected colonies were sectioned and analysed, for both males and females in all sampled months (with the exception of October for male colonies, as only two colonies were available, see Table 1). For male polyps, mesenteries were sectioned in serial slides (4 µm in thickness), whereas mesenteries of female polyps were serially sectioned (4 µm in thickness) every 50 to 80 µm, depending on the monthly average diameter of oocyte nucleus. Sections were performed with a HM 330 Microm rotary microtome and each slide was examined using an Olympus (BX51) compound microscope. In order to avoid counting and measuring the same oocyte twice, the method used by Waller et al. (2014) was applied by counting only those oocytes which had their nucleus cut (a total of 361 for February, 386 for May, 302 for July and 207 for October). Maximum and minimum diameter (µm) were measured for each oocyte. Measurements were performed with the imaging software Cell^D (Olympus Europe), connected to an Olympus DP 20 camera.

Reproductive cycle and reproductive output

Gametogenic stages of development were established using a four stage scale of gamete maturation, adapted by the authors to the analysed species (Table 2) and following the criteria used in previous work (Feehan, Waller & Häussermann, 2019; Mercier & Hamel, 2010; Waller, Tyler & Gage, 2002; Waller & Tyler, 2005). For male polyps, 100 spermatic cysts selected haphazardly from each of the three haphazardly selected mesenteries were analysed and classified following the maturation stages described in Table 2. For female polyps, the total number of oocytes in each of the three haphazardly selected mesenteries were counted and classified in maturation stages following the classification of Table 2. Furthermore, in order to describe the gametogenic cycle (developmental stage), the minimum and maximum diameter (as described in the paragraph above) of 100 oocytes haphazardly selected were measured.

Table 2 Gamete maturation stages for females and males of Dendrophyllia ramea.

Gamete	Maturation stage	Characteristics of the different maturation stages	
Oocyte	Stage I	Previtellogenic oocytes—small oocytes (<42 μm) with basophilic cytoplasm.	
	Stage II	Early vitellogenic oocytes (42–160 μm)—small yolk granules visible in the cytoplasm. Cortical granular layer surrounding oocyte absent or incompletely formed.	
	Stage III	Late vitellogenic oocytes (160–350 μm)—characterized by displacement of the nucleus from its central position. Abundant yolk granules in the cytoplasm cortical granular layer appear fully defined.	
	Stage IV	Mature oocytes >350 μm in mean diameter. The nucleus is totally displaced towards the animal pole.	
Spermatocytes	Stage I	Early—only loosely packed spermatocytes could be seen.	
	Stage II	Maturing—spermatocytes and spermatids are observed in a centripetal maturation gradient. Only some spermatozoa tails are visible in the lumen.	
	Stage III	Mature—lumen filled with densely packed spermatozoa, with the presence of spermatids in the periphery of the spermatocyst.	
	Stage IV	Spent—only relict spermatozoa inside the spermatocyst.	
Note:

Maturation stages display in the table have been modified and adapted from previous studies by Feehan, Waller & Häussermann (2019), Mercier & Hamel (2011), Waller, Tyler & Smith (2008), Waller & Tyler (2005).

Fecundity

The fecundity of D. ramea was calculated following the methodology outlined by Mercier & Hamel (2011). The authors distinguished between potential relative fecundity (PRF), defined as the total number of oocytes per polyp irrespective of their maturity stage, and effective relative fecundity (ERF), defined as the number of mature entities (Stage IV, see Table 2) per polyp. To determine PRF and ERF the total number of oocytes was quantified in three haphazardly selected mesenteries per polyp and colony, which were then averaged. The number of oocytes per mesentery was multiplied by the mean number of pairs of mesenteries per polyp to obtain the fecundity per polyp.

Environmental factors

In order to explore the potential triggers of D. ramea reproductive timing and gamete or larval release, data of sea surface temperature (SST) and primary production (Chl-a) was obtained from open access oceanographic datasets. Monthly Sea surface temperature (SST) data for the years 2017 and 2018 corresponding to the Málaga buoy (the closest buoy to the SAC, 32 nautical miles from the sampling location (36° 41′N 004° 24′W)) were obtained from the Ministerio de Fomento de España (https://www.puertos.es/es-es). The monthly SST data was downloaded from the Historical Data Section and the measuring station (Málaga buoy), whereas monthly Chl-a data from the Alborán Sea were obtained from the Andalusian Environmental Information Network (www.juntadeandalucia.es/medioambiente/site/rediam), specifically from the historical series 2000–2018 available on the Rediam portal. The values of both SST and Chl-a per month correspond to the average of each year during this 2 year period.

Data analysis

A chi-square test was performed to determine whether the sex ratio was significantly different from 1:1. Due to the lack of normality and variance homogeneity in the distribution of the oocyte measurements, the non-parametric Kruskal-Wallis test with a Dunn post hoc test were applied to assess potential differences in average oocyte-size among months and year seasons. Potential differences in oocyte numbers (PRF) were assessed with a one-way ANOVA, after verifying the normality and the variance homogeneity, while the Kruskal-Wallis test with a Dunn post hoc test were used to compare the monthly ERF. A linear regression was used to determine the potential correlation between polyp height (H) and diameter (D) with fecundity (PRF and ERF). All results are presented as mean ± standard deviation (SD).

Results

Morphology

Polyp diameters ranged from 6 to 14 mm, whereas polyp heights ranged from 6 to 37 mm (Supplemental Material 1). The dissection of mesenteries of five haphazardly selected polyps reveal that D. ramea polyps hold an average of 20 ± 4 pairs of mesenteries per polyp. No external morphological differences were detected between males and females.

Sexual pattern and reproductive mode

Dendrophyllia ramea is a gonochoric species. No hermaphroditic specimens were found in this study. In all months sampled, colonies with gametes were observed. Out of 38 colonies examined, 17 (45%) were males, 16 (42%) were females, and the remaining five (13%) did not contain gametes, hence their sex could not be determined. Therefore, D. ramea displays a sex ratio of approximately 1:1 (X2 = 0.03, P = 0.861). Spermatocysts and oocytes are embedded in the mesenteries, surrounded by the mesenterial filament. Fertilization in this species is strongly suggestive of being external, given the absence of larva and/or embryos in any of the histological slides subjected to analysis.

Gametogenesis and reproductive periodicity

As the sampling took place across two consecutive years (2017 and 2018) in order to present the gametogenic cycle following the natural sequence of the months of the year, results have been arranged as follows: February 2018, May 2017, July 2018, October 2017.

Spermatogenesis

According to our classification (Table 2), Stage I (SI) spermatocysts (spermaries) were observed in February (Fig. 3A) and in May (Fig. 3B) with 100% of male colonies displaying this developmental stage (Fig. 4A). In July (Figs. 3C–3E), spermatocysts in SII and SIII were first observed, with SIII being the most abundant developmental stage (37%), followed by SI (26%), SII (23%) and SIV (14%) (Fig. 4A). In October (Fig. 3F) most of the spermatocysts were in SIV (78%), followed by a smaller proportion in SIII (19%) and a minority in SI (3%) (Fig. 4A).

Figure 3 Histological sections of Dendrophyllia ramea reproductive tissues of a male colony.

(A) February 2018, SI = Stage I; (B) May 2017, SI = Stage I; (C and E) July 2018, SI, II, III and IV = Stages I, II, III and IV; (F) October 2017, SIV = Stage IV. st = spermatocyte, sd = spermatid, sz = spermatozoa. Scale bars: A, B, D and F = 50 µm; C and E = 100 µm. Note the results have been arranged by month, although the sampling year was different.

Figure 4 Proportion of the different stages of development of spermatocysts and oocytes of Dendrophyllia ramea.

(A) Spermatocysts (100 spermatocysts) and (B) oocytes for each sampled month (361 oocytes in February 2018, 386 in May 2017, 302 in July 2018 and 207 in October 2017). Spermatocysts and oocytes were analysed from three colonies for each sampled month and each sex, with the exception of October where only two male colonies were available. Note the results have been arranged by month, although the sampling year was different.

Oogenesis

According to our classification (Table 2) mature oocytes (Fig. 5E) are observed in all sampled months. Previtellogenic oocytes (SI; Figs. 5A and 5G) were only visible in February and October, representing 7% and 1% of the observed oocytes, respectively (Fig. 4B). In May (Figs. 5C and 5D), SIII and SIV were the most abundant stages (45% and 33%, respectively, Fig. 4B), with the presence of SII in a lower proportion (22%) and absence of SI. In July (Figs. 5E and 5F), late vitellogenic oocytes SIII (Figs. 5D and 5H) dominated (42%), with high presence of mature oocytes SIV (40%) and some early vitellogenic oocytes SII (Fig. 5B) (18%) and none in SI. Mature oocytes SIV (Fig. 5E) decreased in October (3%) where oocytes in SIII and II dominate (51% and 44% respectively), with almost no oocytes in SI (2%).

Figure 5 Histological sections of Dendrophyllia ramea reproductive tissues of a female colony.

(A and B) February 2018, SI and S II; (C and D): May 2017, S II, S III, SIV; E and F: July 2018, S III and SIV; (G and H) October 2017, SI and SIII. pv, previtellogenic oocyte; evo, early vitellogenic oocyte; lvo, late vitellogenic oocyte; mo, mature oocyte; fc, follicular cells; cr, chorion; y, vitellum; N, nucleus; n, nucleolus. Scale: c, e = 200 µm, d, h = 100 µm, a, b = 50 µm, f, g = 20 µm. Note the migrated nucleolus (pointed out with an arrow) in panel. Note the results have been arranged by month, although the sampling year was different.

The maturity stages of oocytes are also related to different oocyte diameters, with the latter displaying significant differences between the months (H = 277.69, P = 2.20 e-16; Fig. 6). Oocyte diameter ranged from a minimum size of 14.41 μm detected in February to a maximum size of 642.71 μm detected in July. In February the average oocyte diameter was 154.0 μm ± 99.9, which increased substantially in May (277.0 μm ± 127.0) and reached its maximum in July, with a mean oocyte diameter of 302.4 μm ± 141.1. A significant decrease in oocyte diameter was observed in October (184.8 μm ± 89.5, P < 0.001). Although there were no significant differences between the oocyte average diameter in May and July (Fig. 6, Table 3, P = 0.0742), the oocyte size-frequency distributions (Fig. 7) show two different modes for oocyte sizes in February, May and July, while only one oocyte cohort was observed in October, corresponding to small oocytes SI. In all sampled months D. ramea polyps presented a standing stock of small oocytes (SI or SII) in their mesenteries, indicating overlapping gametogenic cycles or a continuous production punctuated by periods of rapid maturity. This pattern strongly indicates the species as iteroparous.

Figure 6 Oocyte size (mean ± SD) for the three female colonies analyzed per sampling month.

Number of measured oocytes was 361 oocytes for February 2018, 386 for May 2017, 302 for July 2018 and 207 for October 2017.

Table 3 Results of the Kruskal-Wallis and post hoc Dunn tests, for the average oocyte diameter per month.

Kruskal-Wallis				
H	P			
277.69	2.2e–16			
Dunn post hoc				
	October	July	May	
February	0.0512	2.2e–16	2.2e–16	
May	2.2e–14	0.0742		
July	2.2e–16			

Figure 7 Oocyte mean diameter (±SD) size-frequency distributions for D. ramea in all the sampled months.

N, colony number; n, oocyte number. Dashed lines indicate the two cohorts 1 (black) and 2 (grey). Note that May and October samples were collected in 2017 and February and July samples were collected in 2018.

Fecundity

The highest average PRF was detected in May (574.7 ± 245.0 oocytes per polyp (opp)) (Figs. 8A and 8B) and the lowest in October (308.2 ± 134.2 opp), decreasing by 46.4% from May to October (Figs. 8A and 8B). However, no statistically significant differences in average PRF among the 4 months have been detected (F = 2.67, P = 0.064). Statistically significant differences were detected among months for the average effective relative fecundity (ERF, oocytes > 350 μm) (K = 17.60, P = 0.00053). As for PRF, the highest average ERF was in May (187.6 ± 81.9 opp) and the lowest in October (10.4 ± 5.2 opp), with a significant decrease of 94.4 5% (Figs. 8A and 8C). The average in February (20.8 ± 25.4 opp) was significantly lower compared to May and July (187.6 ± 81.9 opp and 181.6 ± 90.3 opp) (Figs. 7A and 7C). There was no statistically significant correlation between polyp height or polyp calyx diameter and the fecundity, either PRF or ERF (R2 < 0.1 and P > 0.05 for all combinations; PRF-D: R2 = −0.026 and P = 0.741; PRF-H: R2 = 0.045 and P = 0.113; ERF-D: R2 = −0.002 and P = 0.338; ERF-H: R2 = −0.018 and P = 0.543; D = polyp calyx diameter and H = polyp height).

Figure 8 Mean (±SD) Potential Relative fecundity (PRF, grey line) and Mean (±SD) Effective Relative fecundity (ERF, black line) for D. ramea for each sampling month.

(A) Mean (±SD) Potential Relative fecundity (PRF, grey line) and Mean (±SD) Effective Relative fecundity (ERF, black line) for D. ramea for each sampling month, (B) PRF for each analyzed colony (each analysed colony is indicated in the X axis), (C) ERF for each analyzed colony. Note that May and October samples were collected in 2017 and February and July samples were collected in 2018.

Environmental factors

The decrease in the number of mature oocytes in October 2017 (see Figs. 7 and 8) coincides with lower values of chlorophyll concentration and relatively higher temperatures (Fig. 9). Concentration of Chl-a increased from October to April, with a maximum between March and April, whereas from March to August SST increases, with the highest values occurring in August. Consequently, the months characterised by high ERF (May, July) correspond with periods of lower Chl-a values and rising temperatures.

Figure 9 Potential relative fecundity (PRF) and Effective relative fecundity (ERF) of D. ramea for the samples months, against Chl-a and Sea Surface Temperature values.

(A) Potential relative fecundity (PRF) and Effective relative fecundity (ERF) for the samples months and Chl-a, and (B) Potential relative fecundity (PRF) and Effective relative fecundity (ERF) for the samples months and SST. Note that May and October samples were collected in 2017 and February and July samples were collected in 2018.

Discussion

The results acquired suggest that Dendrophyllia ramea is a gonochoric species. This inference is drawn from the fact that all colonies examined displayed a single sex and no hermaphrodites were observed. Although according to Harrison (2011) gonochorism is not dominant among scleractinians (26% out of the 416 species), it is important to note that most species considered in that study originate from shallow waters, where hermaphroditism prevails. Instead, gonochorism seems to dominate amongst CWC scleractinian species, as 80% of all studied species have been observed to present this reproductive mode (Table 4). To date, only three solitary CWC belonging to the genus Caryophyllia have been documented as hermaphrodites ( Waller & Tyler, 2005). However, an exception to this trend was documented by Pires, Silva & Bastos (2014) who found some specimens of the colonial corals M. oculata and L. pertusa with hermaphroditism patterns. However, the overall sexual pattern in scleractinian corals is highly stable within taxonomic groups, from family to species level (Baird, Guest & Willis, 2009; Harrison & Wallace, 1990; Kerr, Baird & Hughes, 2011). This is also the case of the family Dendrophyllidae, in which 80% of the investigated species, including D. ramea, are gonochoric (Table 5). It is important to keep in mind that, while D. ramea is known to inhabit depths exceeding 150 m in many of its documented occurrences, the results from this study have been obtained from colonies collected at ~40 m. Reproductive features of corals can change across bathymetric ranges as well as geographically (Baird, Guest & Willis, 2009), and differences in reproductive patterns have been found—for instance—for Lophelia pertusa in different areas (Brooke & Järnegren, 2013).

Table 4 Reproductive characteristics of cold-water corals.

Species	Area	Sampling depth (m)	Species depth range (m)	Sex. pattern	Rep.mode	Max oocyte diameter (µm)	Fecundity (ERF) (opp or oo cm2)	Rep. strategy	Time spawning	Refs.	
Caryophyllia cornuformis
S	NE Atlantic (Porcupine Seabight)	1,650–2,017	435–2,000	h	bs	350	?	Quasi-continuous	?	Waller (2005)	
Caryophyllia ambrosia
S	NE Atlantic (Porcupine Seabight)	2,315–2,713	1,100– 3,000	h	bs	700	200–2750 opp	Quasi-continuous	?	“	
Caryophyllia seguenzae
S	NE Atlantic (Porcupine Seabight)	1,240–1,404	960–1,900	h	bs	450	52–940 opp	Quasi-continuous	?	“	
Flabellum alabastrum
S	NE Atlantic (Rockall Trough)	170–2,190	357–2,000	g	bs	925	2800 opp (monthly average fecundity)	Quasi-continuous	?	Waller & Tyler (2011)	
Flabellum angulare
S	NW Atlantic (Canada)	925–1,430	900–3,186	g	bs	1,200	1800–10000 opp	Seasonal	Apr–Jun (spring-summer)	Mercier & Hamel (2011)	
“	NE Atlantic (Porcupine Seabight)	2,412–2,467	900–3,186	g	bs	1,015	550 opp (max average fecundity March)	Seasonal/ periodic	Aug–Sep (late summer)	Waller & Tyler (2011)	
Flabellum curvatum
S	Antarctica (western Antarctic Peninsula)	500–700	115–1,137	g	b	5,120	1618 ± 1071 opp	?	?	Waller, Tyler & Smith (2008)	
Flabellum impensum
S	Antarctica (western Antarctic Peninsula)	270–300	46–2,270	g	b	5,200	1270 ± 884 opp	?	?	“	
Flabellum thouarsii
S	Antarctica (western Antarctic Peninsula)	270–650	71–600	g	b	4,800	2412 ± 1554 opp	?	?	“	
Fungiacyathus marenzelleri
S	NE Atlantic (Rockall Trough)	2,200	300–6,238	g	bs	750	2892 ± 44.4 opp	Quasi-continuous	Jun–Jul (summer)	Waller et al. (2023)	
“	NE Pacific (California)	4,100	300–6,238	g	bs	750	1290 ± 407 opp	Quasi-continuous	?	Flint, Waller & Tyler (2007)	
“	Antarctica (western Antarctic Peninsula)	520–800	300–6,238	g	bs	1,400	2837 ± 121 opp	Quasi-continuous	?	Waller & Feehan (2013)	
Dendrophyllia ramea
C	W Mediterranean (Alborán Sea)	33–37	40–150	g	bs	450	187.6 ± 81.9 opp	Seasonal	August–Sep (late summer)	This study	
Desmophyllum dianthus
C	SE Pacific (Patagonian fjords)	18–27	8–2,500	g	bs	380	2448–172328 opp*	Seasonal	Aug–Sep (late winter)	Feehan, Waller & Häussermann (2019)	
Enallopsammia rostrata
C	SW Pacific (New Zealand)	890–1,130	110–2,165	g	bs	400	>144 opp	Continuous	Apr–May (autumn)	Burgess & Babcock (2005)	
“	SWAtlantic (Brazil)	565–639	110–2,165	g	bs	1,095	?	Continuous	?	Pires, Silva & Bastos (2014)	
Goniocorella dumosa
S	SW Pacific (New Zealand)	890–1,130	88–1,488	g	bs	135	>480 opp	Seasonal	Apr–May (autumn)	Burgess & Babcock (2005)	
Lophelia pertusa
C	NE Atlantic (Porcupine Seabight)	785–980	10–-2,000	g	bs	140	3146–3327 oo cm2	Seasonal	Jan–Feb (winter)	Waller & Tyler (2005)	
“	NE Atlantic (Norway, Trondheim fjord)	40–500	100–2,000	g	bs	180	?	Seasonal	Jan–Mar (winter-spring)	Brooke & Järnegren (2013)	
“	SW Atlantic (Brazil)	565–639	100–2,000	g	bs	242	?	Seasonal /periodic	May–Jul (autumn-winter)	Pires, Silva & Bastos (2014)	
Madrepora oculata
C	NE Atlantic (Porcupine Seabight)	870–925	50–3,600	g	bs	350	10–68 opp/256 oo cm2	Periodic	?	Waller & Tyler (2005)	
“	SW Pacific (New Zealand)	890–1,130	50–3,600	g	bs	?	?	?	?	Burgess & Babcock (2005)	
“	SW Atlantic (Brazil)	565–639	50–3,600	g	bs	650	?	Continuous	?	Pires, Silva & Bastos (2014)	
Oculina varicosa
C	NW Atlantic (Florida)	80–100	3–100	g	bs	100	2115–4693 oo cm2	Periodic	Aug–Sep (late summer)	Brooke & Young (2003)	
Solenosmilia variabilis
C	SW Pacific (New Zealand)	890–1,130	220–2,165	g	bs	165	>290 opp	Seasonal	Apr–May (autumn)	Burgess & Babcock (2005)	
“	SW Atlantic (Brazil)	565–639	220–2,165	g	bs	337	?	Continuous	?	Pires, Silva & Bastos (2014)	
Note:

The table is arranged after solitary (S) and colonial (C) species and following the alphabetical order in each category. h, hermaphrodite; g, gonochoric; bs ,broadcaster spawner; b, brooder; opp, oocytes per polyp; oo cm2, oocytes per cm2; ERF, effective relative fecundity; *PRF, potential relative fecundity.

Table 5 Reproductive strategies of the species belonging to the Dendrophyllidae family (modified from Goffredo et al., 2010).

Species	Sexual pattern	Reproductive mode	References	
Astroides calycularis	g	b	Goffredo et al. (2010, 2011), Terrón-Sigler (2016)	
Astroides calycularis	h	b	Lacaze-Duthiers (1873)	
Balanophyllia elegans	g	b	Beauchamp (1993), Fadlallah (1981, 1983b), Fadlallah & Pearse (1982)	
Balanophyllia europaea	h	b	Goffredo, Telò & Scanabissi (2000), Goffredo, Arnone & Zaccanti (2002), Goffredo & Telo (1998)	
Balanophyllia regia	?	b	Fadlallah (1983a), Kinchington (1981), de Lacaze-Duthiers (1897), Lyons (1973), Yonge (1932)	
Balanophyllia sp.	?	b	Abe (1937), Fadlallah (1983a), Richmond & Hunter (1990)	
Cladopsammia rolandi	h	b	Fadlallah (1983a), de Lacaze-Duthiers (1897)	
Cladopsammia gracilis	?	b	Hizi-Degany et al. (2007)	
Dendrophyllia manni	?	b	Edmondson (1929, 1946), Fadlallah (1983a), Richmond & Hunter (1990)	
Dendrophyllia ramea	g	bs	This study	
Dendrophyllia sp.	g	b	Babcock et al. (1986), Richmond & Hunter (1990)	
Heteropsammia aequicostatus	g	bs	Harriott (1983), Richmond & Hunter (1990)	
Heteropsammia cochlea	g	bs	Harriott (1983), Richmond & Hunter (1990)	
Leptopsammia pruvoti	g	b	Goffredo et al. (2005), de Lacaze-Duthiers (1897)	
Rhizopsammia minuta	?	b	Abe (1937), Fadlallah (1983a)	
Stephanophyllia formosissima	?	b	Fadlallah (1983a),
Moseley (1881)	
Tubastrea aurea	?	b	Edmondson (1929, 1946), Fadlallah (1983a), Fan et al. (2006)	
Tubastrea coccinea	h	b	Creed & De Paula (2007), Edmondson (1929, 1946), Glynn et al. (2008), Richmond & Hunter (1990), Jokiel, Ito & Liu (1985), Petersen et al. (2007)	
Tubastrea faulkneri	g	b	Babcock et al. (1986), Richmond & Hunter (1990)	
Tubastrea tagusensis	?	b	Creed & De Paula (2007)	
Turbinaria bifrons	?	bs	Babcock, Wills & Simpson (1994)	
Turbinaria frondens	g	bs	Babcock, Wills & Simpson (1994), Richmond & Hunter (1990), Willis et al. (1985), Wilson & Harrison (2003)	
Turbinaria mesenterina	?	bs	Babcock, Wills & Simpson (1994)	
Turbinaria peltata	?	bs	Babcock, Wills & Simpson (1994)	
Turbinaria radicalis	?	bs	Babcock, Wills & Simpson (1994), Wilson & Harrison (2003)	
Turbinaria reniformis	g	bs	Babcock, Wills & Simpson (1994), Petersen et al. (2007), Richmond & Hunter (1990), Willis et al. (1985)	
Note:

h, hermaphrodite; g, gonochoric; ?, unknown; b, brooder; bs, broadcaster spawner.

Regarding the reproductive mode of D. ramea, the absence of larvae and/or embryos in the analysed colonies suggests that this species is a broadcast spawner (i.e., releases the gametes to the water column). This seems to be the prevailing reproductive mode in CWCs; only three of the CWCs species investigated to date are brooders, all of which belong to the genus Flabellum: F. impensum, F. curvatum and F. thouarsii (Waller, Tyler & Smith, 2008) (Table 4). However, the reproductive strategy is more variable within the same scleractinian taxon (i.e., family, genus, species) than the sexual pattern (Harrison & Wallace, 1990, Kerr, Baird & Hughes, 2011). An illustrative case can be observed in the genus Porites from shallow tropical waters, which includes 10 brooder and 10 broadcaster species, while it only presents two gonochoric species in the Atlantic and the Indo-Pacific (Baird, Guest & Willis, 2009). Nevertheless, most of the species of the family Dendrophyllidae are brooders, with eight broadcast spawner species belonging to the genus Heteropsammia and Turbinaria (Table 5).

The two oocyte cohorts detected simultaneously in D. ramea, and the fact that mature oocytes -with migrated nucleus- can be found in July (see Fig. 5F), suggest a yearlong gametogenic cycle, characterised by a prolonged oocyte maturation and a seasonal spawning occurring from August to October. One of the most striking observations in this study was the absence of initial developmental stages in females (oogonia) and males (spermatogonia). The most plausible explanation for this is the fact that our observations have been focused on the mesenterial mesoglea. Several authors have documented the migration of gametes for shallow water corals in early developmental stages (Stage 0) from the gastrodermis to the mesenterial mesoglea, where differentiation and subsequent maturation takes place (Fadlallah, 1983a; Goffredo, Arnone & Zaccanti, 2002, 2004, 2012; Szmant-Froelich, Yevich & Pilson, 1980). The lack of occurrence of early oocyte stages in the mesenteries has been also documented in the CWCs Fungiacyathus marenzelleri (Waller, Tyler & Gage, 2002), Lophelia pertusa and Madrepora oculata (Waller & Tyler, 2005), Caryophyllia sp. (Waller, Tyler & Gage, 2005) and Flabellum sp. (Waller & Tyler, 2011). Therefore, we suggest that the migration from gastrodermis to mesenterial mesoglea at the beginning of the gametogenesis also occurs in D. ramea. However, further investigations should add specific analyses of the gastrodermis in order to be able to potentially detect oogonia and spermatogonia.

Regarding the developmental stages for oocytes, there is a wide range of maturation scales, proposed by different authors. For instance, Waller & Feehan (2013) consider only previtellogenic and vitellogenic oocytes to describe the gametogenesis of F. marenzelleri, whereas in previous works (Waller, Tyler & Gage, 2002; Flint, Waller & Tyler, 2007), an additional stage was considered: late-vitellogenic. In our study, we consider four stages (excluding oogonia due to their absence), since clear differences have been observed amongst them (see Fig. 5).

We also described four developmental stages for spermatogenesis, following the criteria applied for the CWC species F. mazzerelli (Waller, Tyler & Gage, 2002), L. pertusa (Brooke & Järnegren, 2013) and Caryophyllia sp. ( Waller & Tyler, 2005). The spermatogenesis stages of D. ramea are also similar to other Mediterranean corals, although the latter include on additional stage: spermatogonia, resulting in a total of five stages. This applies to L. pruvoti, A. calycularis and C. inornata (Goffredo, Arnone & Zaccanti, 2002, 2010, 2012).

Within scleractinian CWCs and DSC there are some species that reproduce seasonally whereas others showcase patterns of continuous or quasi continuous reproduction (Waller, 2005). Dendrophyllia ramea seems to have a seasonal reproductive pattern with gamete release occurring between August and October (Figs. 5 and 7). The absence of the larger oocyte cohort in October, together with the high percentage of spermatocysts in SIV (spent stage), suggest that gamete release most probably occurs in the period from the end of summer and to the start of autumn. Furthermore, the finding of late vitellogenic oocytes of ~200–300 µm, could suggest (following Strathmann, 1978) a pelagic or a demersal development of larvae.

Our results for D. ramea showing different oocyte developmental stages simultaneously, suggest two possible reproductive strategies: (1) a quasi-continuous gametogenesis or, (2) a gametogenesis extended over time with a periodic gamete release. The latter option seems to be the most probable considering that: (1) our results show significant differences in average oocyte size within each month compared to other species with quasi-continuous reproduction, which do not show significant differences among oocyte sizes (Flint, Waller & Tyler, 2007; Waller, Tyler & Gage, 2002), and (2) due to the low number of previtellogenic oocytes (see Fig. 4), as quasi-continuous reproduction implies a larger number of previtellogenic oocytes. A periodic gamete release over time has been previously suggested for other CWC species (Waller & Feehan, 2013); these authors also suggested the two potential strategies discussed earlier (i.e., quasi-continuous gametogenesis and long term gametogenesis with periodic gamete release) for the Antarctic solitary coral F. marenzelleri. Burgess & Babcock (2005) suggested that the simultaneous presence of oocytes SIII and SIV in Enallopsammia rostrata in the Pacific Ocean was due to a delay in oocyte development, which was associated with nutritional resource availability. Periodic gamete releases have been observed for Oculina varicosa in the Atlantic, with up to 31 days of difference between spawning events, which show that gamete release occurs over long periods of time (Brooke & Young, 2003). In the present study, the two oocyte cohorts found in D. ramea, suggest that oocyte maturation may take longer than 12 months. However, as previously mentioned, the low number of previtellogenic oocytes impairs our ability to ascertain the initiation of gametogenesis. Analysis of samples from additional time periods could potentially provide further insights.

The developmental stages of the spermatocysts reveal an annual cycle, shorter than the oogenesis; which is common for anthozoans (Goffredo, Arnone & Zaccanti, 2002; Guest et al., 2005; Harrison & Wallace, 1990; Richmond & Hunter, 1990; Schleyer, Kruger & Benayahu, 2004). Our data suggest a start of the spermatogenesis in the winter time (SI), with increased activity in the summer and release of mature sperm in autumn (Fig. 4). Similar patterns of spermatogenesis have been documented for other CWC and DSC species: L. pertusa (Brooke & Järnegren, 2013; Pires, Silva & Bastos, 2014), M. oculata (Waller & Tyler, 2005) and F. marenzelleri (Flint, Waller & Tyler, 2007; Waller, Tyler & Gage, 2002). Although they inhabit shallower waters, the temperate Mediterranean species B. europaea, L. pruvoti and A. calycularis (brooding species from the family Dendrophyllidae) also present a longer oogenesis than spermatogenesis (Goffredo, Arnone & Zaccanti, 2002, 2010, 2012, 2005, 2011).

The period of gamete release occurs, in the analysed location where D. ramea thrive, when temperature is higher and Chl-a values are low. In the Alborán Sea, the investigated D. ramea population occurs in a depth range from 16 to 50 m. This region is distinctive for its deep water upwellings, which hold significant influence over both temperature dynamics and the productivity patterns of the area (Sarhan et al., 2000). It is known for several coral species, from both shallow and deep waters, that temperature, photoperiod, lunar cycles and food supply are environmental factors influencing reproduction timing (Goffredo et al., 2006, 2011; Glynn et al., 2000; Harrison, 2011; Richmond, 1997; Waller & Tyler, 2005). Indeed, the reproductive cycle of the temperate broadcast spawner corals O. patagonica and C. caespitosa, revealed a strong correlation with temperature in the western Mediterranean. Notably, these species showcase a peak in reproductive activity during the summer months, with gamete release taking place towards the end of summer as temperatures start to decrease (Fine, Zibrowius & Loya, 2001; Kersting et al., 2013). In the present study the maximum temperature peak occurred in August and the gamete release seems to take place between August-September, when temperatures rapidly decrease and the Chl-a values are displaying minimum values. In October, Chl-a concentrations rapidly increase, which could be beneficial for the larvae (if they were to be planktotrophic) by providing better nutritional conditions in the surrounding waters. This increase in Chl-a concentration, which extends until April, could be beneficial for the start of the gametogenesis in the early months of the year, allowing to accumulate reserves for further gamete development. In corals (and other organisms) it becomes evident that the availability of food is the most important trigger influencing the timing of gametogenesis, while other factors, such as temperature, cue spawning (Feehan, Waller & Häussermann, 2019 and references therein).

Dendrophyllia ramea presents relatively low PRF values (PRF máx.: 925 opp). This stands in contrast to other CWCs and DSC, where values are notably higher by an order of magnitude (Table 4). Nevertheles, similar fecundities have been documented for C. seguenzae (in the higher part of the range) and F. angulare in the NE Atlantic (Table 4, Waller & Tyler, 2005; Mercier, Sun & Hamel, 2011 respectively). Some authors suggest that depth can be a factor constraining fecundity (Flint, Waller & Tyler, 2007; Waller, Tyler & Gage, 2002, 2008). However, when comparing different species this is not always the case. For instance, C. seguenzae and F. angulare were sampled at much greater depths (1,240–1,409 m, 925–1,430 m respectively, see Table 4) than D. ramea (33–37 m) in this study, yet they display similar fecundity values. On the contrary, D. ramea and D. dianthus sampled at very similar depths (33-37 m in this study and 18–27 m in Feehan, Waller & Häussermann, 2019 respectively) revealed very different fecundity values, with a much higher gamete production in D. ramea than in D. dianthus (925 opp vs. 2,448–172, 328 opp, see Table 4 and Feehan, Waller & Häussermann, 2019). However, it is worth mentioning that the comparison of fecundity among species is not always possible, as the applied methodologies vary. For instance, (Goffredo, Arnone & Zaccanti, 2002, 2011; Goffredo et al., 2006) calculated fecundity for Mediterranean temperate corals using a formula that considers the length of the “ovary” (which is the mesentery where the oocytes develop) and the frequency and diameter of mature oocytes; other authors calculate the fecundity at colony level (Brooke & Young, 2003). In studies conducted with F. marenzelleri (Flint, Waller & Tyler, 2007; Waller, Tyler & Gage, 2002) and M. oculata (Waller & Tyler, 2005), more than one oocyte cohort have been detected; yet, fecundity was calculated considering the total number of opp (PRF) (Flint, Waller & Tyler, 2007; Waller, Tyler & Gage, 2002; Waller & Tyler, 2005). There is only one study regarding scleractinian CWCs (F. angulare, NW Atlantic), by Mercier & Hamel (2011), in which fecundity was measured by taking into consideration only the mature oocytes (ERF). The study revealed very high values, with a maximum of 10,000 opp, two orders of magnitude higher than the results obtained in our study for D. ramea (ERF max.: 536 opp). Up to date, maximum oocyte size diameter for scleractinian CWCs ranges from 100 μm (Brooke & Young, 2003) to 5,200 μm (Waller, Tyler & Smith, 2008). The maximum oocyte diameter measured for D. ramea (617 μm) is in the middle part of the range for CWC and DSC, as of the 17 species investigated (in some cases the same species was analysed for different areas), 12 species have smaller oocytes, whereas the other 12 have larger oocytes than D. ramea. The largest diameter (4,800–5,167 μm) corresponds to the Antarctic populations of the genus Caryophyllia. This might be related to the large lipid deposits included in the oocytes, a typical adaptation of Cold-water organisms (Waller, Tyler & Smith, 2008).

No relationship has been detected between fecundity and polyp size, considering both polyp calyx diameter and height. In general, the reproductive output is related to body size in most marine invertebrates (Gage & Tyler, 1991; Hall & Hughes, 1996). However, this does not seem to apply to CWC and DSC species, as no correlation has been detected for D. dianthus, L. pertusa, M. oculata and C. ambrosia (Feehan, Waller & Häussermann, 2019; Waller & Tyler, 2005; Waller & Tyler, 2005), yet there are exceptions. For instance, a positive correlation has been found in some solitary CWC species: F. marenzelli (Waller, Tyler & Gage, 2002), F. angulare (Mercier & Hamel, 2011; Waller & Tyler, 2011) and F. alabastrum (Waller & Tyler, 2011), which suggests that this aspect of CWC reproduction needs to be investigated further.

Conclusions

This is one of the first investigations dedicated to describe the reproductive characteristics and gametogenic cycle of a colonial azooxanthellate mesophotic and DSC species from the Mediterranean: Dendrophyllia ramea. This coral is gonochoric and the absence of larva suggests a broadcasting reproductive strategy. Dendrophyllia ramea presents two oocyte cohorts in winter, spring and early summer months, whereas in October (autumn) a single oocyte cohort is detected in the polyps, suggesting a seasonal reproduction with spawning taking place in late summer/early autumn. Although the currently available data does not allow for definite proof, the beginning of the oogenesis and spawning season seem to be related to the higher Chl-a values. This correlation could potentially be advantageous for coral colonies, as higher Chl-a values suggest a higher availability of food resources. This is indeed, a factor of paramount importance to promote the energetically costly process of the gametogenesis and larvae production. Knowledge on the reproduction of CWCs is still scarce, but absolutely necessary to improve the understanding of the functionality of these organisms and population dynamics, as well as to design any potential protection and restoration measures.

Supplemental Information

Supplemental Information 1 Collected samples of Dendrophyllia ramea.

The year season of each sample, specific sampling date, location, depth range and number of collected colonies.

Click here for additional data file.

The authors are grateful to Beatriz Torres Hansjosten and Heather Baxter for their help preparing coral samples for the histological study. We are indebted to the divers David León, Patricio Peñalver y Alejandro Ibáñez from the Hombre y Territorio Foundation, who participated in the sampling of the specimens of D. ramea. This manuscript has been substantially improved thanks to the comments and suggestions of Guillem Corbera, Andrea Gori and an anonymous reviewer. The English language of the manuscript has improved thanks to contributions of Sonja Böske da Costa.

Additional Information and Declarations

Competing Interests

Author Contributions

Field Study Permissions

Data Availability

The authors declare that they have no competing interests

Covadonga Orejas conceived and designed the experiments, performed the experiments, analyzed the data, prepared figures and/or tables, authored or reviewed drafts of the article, and approved the final draft.

Silvia Antón-Sempere performed the experiments, analyzed the data, prepared figures and/or tables, authored or reviewed drafts of the article, and approved the final draft.

Alexis Terrón-Sigler performed the experiments, prepared figures and/or tables, and approved the final draft.

Amalia Grau conceived and designed the experiments, performed the experiments, analyzed the data, authored or reviewed drafts of the article, and approved the final draft.

The following information was supplied relating to field study approvals (i.e., approving body and any reference numbers):

Sampling of coral specimens was issued by the Junta de Andalucía (Spain).

The following information was supplied regarding data availability:

The data is available at PANGAEA: Orejas, Covadonga; Antón-Sempere, Silvia; Terrón-Singler, Alexis; Grau, Amalia (2023): Reproductive features and gametogenesis of Dendrophyllia ramea from the Alboran Sea, Western Mediterranean. PANGAEA, https://doi.org/10.1594/PANGAEA.957212.

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
