# Peer review of "Reproductive characteristics and gametogenic cycle of the scleractinian coral Dendrophyllia ramea"

_PeerJ, doi:10.7717/peerj.16079_

## Round 0.1 · original submission · Minor Revisions

We have received evaluations from two expert reviewers. Both agree that research on cold water and deep sea corals is generally lacking so this study does fill a void. Both reviewers have pointed out a long list of minor issues that need to be attended to in order to give this manuscript further consideration. Please ensure that you respond to each point in a rebuttal. Also please note that reviewer 1 refers to two documents. We have merged them to form one PDF for ease. Make sure to go over each comment in the review and those comments embedded within the manuscript.

Reviewer 1 ·

Basic reporting

Information on reproduction of mesophotic/deep-sea/cold-water corals has expanded over the past decade but is still lacking for many of these relatively inaccessible species. The information in the submitted manuscript is therefore worthy of publication. There are no major problems with the work, which is a straightforward analysis of reproduction, but there are a number of inconsistencies and other issues that should be fixed before the manuscript is accepted. I have made observations on each section below but embedded suggested text changes and comments in the PDF, which I have attached to my review. There are grammatical errors throughout the manuscript, and I have corrected many of them but I recommend review by a fluent English speaker prior to resubmission.

Experimental design

The sampling was limited but this is not unusual for species that are difficult to sample. The paper meets PeerJ standards

Validity of the findings

Findings valid. Additional comments in attachments

Additional comments

I have attached a PDF files; one is a review by section with suggested improvements and requests for clarification etc. The other has comments and suggested edits embedded. The two documents include different information and should be reviewed separately.

Annotated reviews are not available for download in order to protect the identity of reviewers who chose to remain anonymous.

·

Basic reporting

This manuscript reports first data on the reproduction of the scleractinian coral Dendrophyllia ramea from the Mediterranean Sea. I just have a general and some minor comments, which I hope can help authors to further improve their study.

General comment

The study is basically introduced and discussed in the frame of cold-water and deep-sea corals ecology. However, this is strictly not the case for the studied species which, as its distribution shows, is present from shallow warm to deep cold waters. Authors should clearly state this and modify the introduction and the discussion accordingly. Moreover, in the discussion it should be clearly stated that the observed reproductive seasonal pattern is for the species in shallow waters, whereas it could be not representative for the species in deeper bottoms in the Mediterranean Sea or the Atlantic Ocean.

Experimental design

The study is new, interesting, and well performed, producing and reporting new data on basic biological processes of a marine species.

Validity of the findings

Same as before.

Additional comments

Minor comments

Line 29 – I would suggest changing “Chla” to “Chl a” all over the text.

Line 35 – I strongly suggest speaking about two cohorts instead two size classes (indeed, oocytes are divided in 13 size classes in figure 6) all over the text.

Line 39 – Change “are” to “to be”.

Line 48 – Change to “Cold-Water Coral (CWC)”.

Line 52 – Use the already defined acronym CWC.

Line 58 – I would say “mainly due to better accessibility”.

Line 61 – I suggest changing “adaptation” to “reproductive pattern”.

Lines 70-76 – Pay attention that the listed species are basically not CWC.

Line 77 – I do not agree with the definition of Leptosamia pruvoti as deep-sea species since it can be commonly found with high abundances from very shallow waters. Please rephrase this sentence also considering the formal definition of deep-sea species as occurring below 200 m depth (not 50 m depth).

Line 100 – Change “deep sea” to “deep-sea”.

Line 108 – Change “nutritive” to “nutrient-rich”.

Line 117 – Change “too” to “took”.

Line 120 – Junta de Andalucia was previously defined as “autonomous Andalusian government”.

Line 130 – Polyp lenght was previously defined (Line 127) as polyp height. Please, chose one.

Line 135 – I suggest changing “have been” to “were”.

Line 151 – Remove the comma after “were”.

Line 152 – I do not understand what “according to the monthly average diameter of oocyte nucleus” means here. Please, clarify.

Line 161 – I suggest rephrasing this sentence avoiding “for us modified” and using something like “adapted to the study species” or similar.

Line 165 – Remove the comma after “mesenteries”.

Line 170 – Delete “using the software Cell^D (Olympus)” as this has already been explained before (Line 158).

Line 184 – Delete “and”.

Line 184 – Add a comma after “datasets”.

Lines 184-185 – Please, add further details about the recordings of SST even if this comes from an external source. Additionally, also add how far is the selected buoy from the sampling site.

Lines 187-188 – Similarly, please add further details about how the Chl a data are recorded and treated, even if this comes from an external source.

Line 206 – Remove “(SD)” all over the result section, since this is already explained before (Line 199).

Line 243 – I strongly suggest indicating the full value of the p only when it is > 0.05, while indicate p-values as < 0.05, < 0,01 or < 0.001 (as in this case) when the difference is statistically significant.

Line 251 – Change “size class” to “cohort”.

Line 257 – Just use PRF, since the acronym has been already defined.

Line 263 – I suggest reducing decimal numbers to just one for most of the reported values, since they basically just derive from a mathematical operation (mean calculation).

Lines 271 – Change “chlorophyll” to “Chl a” as previously defined.

Line 288 and 296 – Change “taxa” (plural) to “taxon” (singular).

Line 301 – Change “size classes” to “cohorts”.

Line 302 – Delete “the” and “samples”.

Line 304 – I suggest changing from “in August, September and October” to “in the period from August to October”.

Lines 304-305 – I suggest the delete this sentence, which is not needed.

Lines 328 and 405 – Delete “temperate” since all shallow Mediterranean corals are temperate.

Line 343 – Delete “(H= 277.69; P<0.001)” that is already clear from the results and is not needed in the discussion.

Line 356 – Change from “size classes” to “cohorts”.

Lines 371-372 – As stated in the conclusion, this study really does not demonstrate the influence of the studied environmental conditions on the release of gametes. It only observes that the release occurs when temperature is higher and Chl a is low. Please, rephrase.

Line 376 – Add temperature among the environmental factors listed here.

Line 386 – Delete a space after “waters”.

Line 391 – Use the already defined acronym PRF.

Lines 398-400 – Add a space from numbers and “m”.

Line402 – Add a space between numbers and “opp”.

Line 414 – Use only the already defined acronym “opp”.

Line 416 – Add a space between “100” and “μm”.

Lines 419 – I would suggest deleting “and included in table 5” that is not needed.

Line 420 – Change “in twelve cases species” to “12 species”.

Line 420 – Delete “cases” after 12.

Line 422 – Change “Genus” to “genus”.

Lines 424-425 – Delete this sentence, which is not needed. Similarly, also delete the starting of the next sentence “However, as explained in the results section”.

Lines 438 – Change “size classes” to “cohorts”.

Lines 439 – Change “size class” to “cohort”.

Table 1 – Change “sampling” to “samples collected”. Delete “(colony/polyp)”. I suggest changing “colony number” to “number of colonies” and “polyp number” to “number of polyps”.

Table 2 – I suggest changing to “male colonies of”.

Table 3 – I suggest changing p-values as explained before (comment to line 243).

Table 5 – Double check the caption, I think the first sentence should be deleted.

Figure 6 – I suggest changing “colony number” to “number of colonies” and “polyp number” to “number of polyps”.

Figure 8 – Change D. ramea to italics.

---

## Round 0.2 · Minor Revisions

Thank you for your detailed rebuttal letter. I have read over the manuscript and am satisfied with the changes made to it. However, there are still some issues with the use of the English language. Please have a fluent English speaker/Writer go over the manuscript one last time. I have indicated some corrections but these are not exhaustive.

**Language Note:** The Academic Editor has identified that the English language must be improved. PeerJ can provide language editing services - please contact us at copyediting@peerj.com for pricing (be sure to provide your manuscript number and title). Alternatively, you should make your own arrangements to improve the language quality and provide details in your response letter. – PeerJ Staff

---

## Round 0.3 · accepted · Accept

I am satisfied with the modifications that you have made to the manuscript.